# MaskFlow: Discrete Flows For Flexible and Efficient Long Video Generation

## Abstract

Generating long, high-quality videos remains a challenge due to the complex interplay of spatial and temporal dynamics and hardware limitations. In this work, we introduce *Mask-Flow*, a unified video generation framework that combines discrete representations with flow-matching to enable efficient generation of high-quality long videos. By leveraging a frame-level masking strategy during training, MaskFlow conditions on previously generated unmasked frames to generate videos with lengths ten times beyond that of the training sequences. MaskFlow does so very efficiently by enabling the use of fast Masked Generative Model (MGM)-style sampling and can be deployed in both fully autoregressive and full-sequence generation modes. We validate the quality of our method on the FaceForensics (FFS) and Deepmind Lab (DMLab) datasets and report Fréchet Video Distance (FVD) competitive with state-of-the-art approaches. We also provide a detailed analysis on the sampling efficiency of our method and demonstrate that MaskFlow can be applied to both timestep-dependent and timestep-independent models in a training-free manner. Code will be released.

## 1 Introduction

Due to the high computational demands of both training and sampling processes, long video generation remains a challenging task in computer vision. Many recent state-of-the-art video generation approaches train on fixed sequence lengths (Blattmann et al., 2023a;b; Ho et al., 2022) and thus struggle to scale to longer sampling horizons. Many use cases also require the ability to generate videos with varying length. A common way to address this is by adopting an autoregressive diffusion approach similar to LLMs (Gao et al., 2024), where videos are generated frame by frame. This approach has other downsides, since it requires traversing the entire denoising chain for every frame, which is computationally expensive. Since autoregressive models condition the generative process recursively on previously generated frames, error accumulation, specifically when generating videos longer than the training videos, is another challenge.

Several recent works (Ruhe et al., 2024; Chen et al., 2024) have attempted to unify the flexibility of autoregressive generation approaches with the advantages of full sequence generation. These approaches are built on the intuition that the data corruption process in diffusion models can serve as an intermediary for injecting temporal inductive bias. Progressively increasing noise schedules (Xie et al., 2024; Ruhe et al., 2024) are an example of a sampling schedule enabled by this paradigm. These works impose monotonically increasing noise schedules w.r.t. frame position in the window during training, limiting their flexibility in interpolating between fully autoregressive and full-sequence generation. This is alleviated in (Chen et al., 2024), where independent noise levels are applied to frames during training, and the diffusion model is trained to denoise arbitrary sequences of noisy frames. All of these works use continuous representations.

We transfer this idea to a discrete token space for two main reasons: First, it allows us to use a masking-based data corruption process, which enables confidence-based heuristic sampling that drastically speeds up the generative process. This becomes especially relevant when considering frame-by-frame autoregressive generation. Second, it allows us to use discrete flow matching dynamics, which provide a more flexible design space and the ability to further increase our sampling speed. Specifically, we adopt a *frame-level*

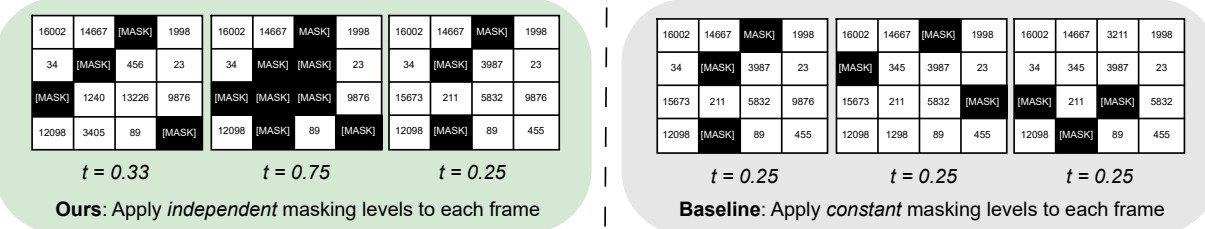

Figure 1: **MaskFlow Training:** For each video, Baseline training applies a single masking ratios to all frames, whereas our method samples masking ratios independently for each frame.

*masking* scheme in training (versus a *constant-level masking* baseline, see Figure 1), where we mask an independently sampled proportion of tokens in each frame in the sequence. This allows us to condition the generative process on an arbitrary number of previously generated frames, while still being consistent with the training task. This makes our method inherently versatile, allowing us to generate videos using both full-sequence and autoregressive frame-by-frame generation, and use different sampling modes. We show that confidence-based masked generative model (MGM) style sampling is uniquely suited to this setting, generating high-quality results with a low number of function evaluations (NFE), and does not degrade quality compared to diffusion-like flow matching (FM)-style sampling that uses larger NFE. Combining frame-level masking during training with MGM-style sampling enables highly efficient long-horizon rollouts of our video generation models beyond $10\times$ training frame lengths without degradation. We also demonstrate that this sampling method can be applied in a timestep-*independent* setting that omits explicit timestep conditioning, even when models were trained in a timestep-dependent manner, which further underlines the flexibility of our approach. In summary, our contributions are the following:

- To the best of our knowledge, we are the first to unify the paradigms of discrete representations in video flow matching with rolling out generative models to generate arbitrary-length videos.

- We introduce MaskFlow, a frame-level masking approach that supports highly flexible sampling methods in a single unified model architecture.

- We demonstrate that MaskFlow with MGM-style sampling generates long videos faster while simultaneously preserving high visual quality.

- Additionally, we demonstrate an additional increase in quality when using full autoregressive generation or partial context guidance combined with MaskFlow for very long sampling horizons.

## 2 Related Work

**Long Video Generation.** The training dynamics and the sampling methodology in this work are inspired by works like Diffusion Forcing (Chen et al., 2024; Song et al., 2025), Rolling Diffusion Models (Ruhe et al., 2024) and AR-Diffusion (Wu et al., 2023). The main motivation behind these works is to unify the benefits of autoregression and full sequence diffusion by applying token-specific noise levels during training, which allows the model to generate future frames without fully denoising past frames in a sequence. Xie et al. (2024) is a similar work that prescribes a progressive sampling schedule for increased smoothness of transitions between generation windows. FIFO-Diffusion is a training-free inference approach for infinite text-to-video generation that uses a similar progressive denoising schedule and latent partitioning to reduce the training-inference gap with pre-trained video diffusion models. Other methods like (Gao et al., 2024; Zheng et al., 2024) and (Blattmann et al., 2023a) use context frame conditioning similar to our method, but do not focus on long video generation. The closest to our work is Zhou et al. (2025), who also employ a masking-based design to generate arbitrary-length videos autoregressively. There are two key differences in our approach: We do not condition frame generation on any previous ground truth frames during training, but adopt a frame-level masking approach that is more flexible. We also employ confidence-based MGM-style sampling, which lets us sample entire training windows in very few sampling steps, whereas Zhou et al. (2025) employs MAR-style

(Li et al., 2024) sampling that requires a higher amount of sampling steps per individual frame and does not use vector quantization.

**Discrete Representations in Video Generation.** There are several previous works that investigate the use of discrete representations for video diffusion. MaskGIT (Chang et al., 2022) is a generative transformer that uses a bidirectional transformer decoder to predict randomly masked tokens in an input sequence of image patches. This idea is extended to videos in MAGVIT (Yu et al., 2023a), which tokenizes video pixel space inputs into spatial-temporal visual tokens and uses a masked auto-regressive approach to predict masked input tokens. Similar approaches like Muse (Chang et al., 2023) and MAGVIT-v2(Yu et al., 2023b) have shown promise in scaling up image and video generation tasks, but suffer from training instabilities.

**Discrete Flow Matching.** Flow matching (Lipman et al., 2023) is an emerging generative modeling paradigm that generalizes common formulations of diffusion models and offers more freedom in the choice of the source distribution. Flow matching models have seen wide adoption in speech (Liu et al., 2023), image generation (Hu et al., 2024a;b; Dao et al., 2023; Lipman et al., 2023), super-resolution (Schusterbauer et al., 2024), depth estimation (Gui et al., 2025) and video generation (Jin et al., 2024), but their application in high-dimensional discrete domains is still limited. Discrete flow matching (Gat et al., 2024; Campbell et al., 2024; Shi et al., 2024; Sahoo et al., 2024) addresses this limitation, introducing a novel discrete flow paradigm designed for discrete data generation. In our work, we explore vectorizing timesteps across frames for memory-efficient long-video generation with improved extrapolation to long sampling horizons while also analyzing the impact of sampling styles on video quality.

## 3 Method

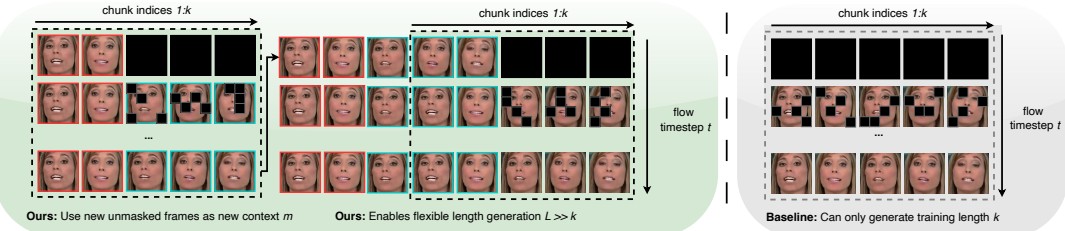

Figure 2: **MaskFlow Sampling:** Given $m = 2$ context frames used to initialize generation, we unmask the current window and use newly generated frames as new context frames in the next chunk of size $k = 5$, using stride $s = 3$. (*Tokenization omitted here to simplify understanding*).

### 3.1 Task formulation: Long video generation

In general, there are three distinct approaches to the generation of long videos. The first is the naive approach of training on longer video sequences. This is challenging due to the quadratic complexity in attention mechanisms with respect to token numbers. Although works like(Tan et al., 2024; Harvey et al., 2022) address this by distributing the generative process or by generating every $n$-th frame and subsequently infilling the remaining frames, the approach remains inefficient. The second approach is a *rolling* (or "sliding-window") approach, which applies monotonically increasing noise dependent on a frame's position in the sliding window. This process can be rolled out indefinitely, removing frames from the window when they are fully denoised and appending random noise frames at the end of the window. Works such as (Ruhe et al., 2024; Wu et al., 2023; Xie et al., 2024) belong to this paradigm. The third approach is *chunkwise-autoregression*, also referred to as blockwise-autoregression (Ruhe et al., 2024). Here, the video of length $L$ is divided into overlapping *chunks* of length $k \ll L$, where each chunk overlaps by $m$ frames, which we refer to as context frames. Concretely, we define a video and its frames as

$$\mathbf{v} = (v^1, v^2, \dots, v^L) \tag{1}$$

which we divide into overlapping chunks of length $k$. Let $\ell = \left\lceil \frac{L-k}{s} \right\rceil + 1$ denote the number of chunks needed to cover the video of length $L$, and we further define each chunk $\mathbf{v}^{(i)}$ as

$$\mathbf{v}^{(i)} = \left(v^{(i-1)\,s+1}, \ldots, v^{(i-1)\,s+k}\right), \tag{2}$$

where $s \leq k$ is the sampling window stride, i.e., how far the context start shifts at each step. Often, one sets $s = k - m$, but this is not strictly required. The video distribution then factorizes as

$$p(\mathbf{v};\theta) \;=\; p(\mathbf{v}^{(1)};\theta) \prod_{i=2}^{\ell} p\left(\mathbf{v}^{(i)} \mid \mathbf{v}^{(i-1)};\theta\right). \tag{3}$$

Because each $\mathbf{v}^{(i)}$ overlaps the previous chunk by $m$ frames, the context frames feed into the next chunk's generation, ensuring smooth transitions and continuity between chunks.

### 3.2 Preliminary: Flow Matching for Videos

Our masking flow matching approach, named *MaskFlow*, draws inspiration from works that apply independent noise levels to individual frames (Chen et al., 2024; Ruhe et al., 2024). These works operate in a continuous space and use diffusion to corrupt data. MaskFlow operates in a discrete token space and uses *masking* to corrupt data. We seek to learn a continuous transition process in "time" $t$ that moves from a purely masked sequence at $t = 0$ to the unmasked token sequence at $t = 1$. In our method, $1 - t$ corresponds to the masking ratio and represents the frame-level probability of a token being masked. Consider a video consisting of $L$ frames, where each frame is mapped to a discrete latent space using a vector-quantized (VQ) tokenizer (Esser et al., 2021). This tokenizer encodes each frame in the video $\mathbf{v}$ to a set of discrete latent indices $\mathbf{x}_{\text{latent}} \in [K]^N$, which consists of $N$ tokens drawn from the tokenizer vocabulary of size $K$. Let $\mathcal{F}$ denote the VQ encoder-decoder, i.e., the function that maps a video in pixel space to its tokenized representation. Then, we have $\mathbf{x} = \mathcal{F}(\mathbf{v}) \in [K]^{L \times N}$, where $[K] = \{1, 2, \ldots, K\}$ is the set of all possible token indices which includes a special "mask token" $M \in [K]$. The choice of tokenization is essential here, since it compresses spatial dimensions of $\mathbf{x}$ compared to $\mathbf{v}$ and allows us to employ discrete flow matching, which we outline in further detail in the following section.

**Discrete Flow Matching.** Discrete flow matching (Gat et al., 2024) defines a vector field $u_t$ in a discrete space that can be traversed to yield a smooth probability transition between our source distribution of fully masked frame sequences $p(\mathbf{x}_0)$ and the distribution of unmasked sequences $p(\mathbf{x}_1)$. This vector field defines an optimal transport path between the two distributions. Concretely, we construct the conditional probability path:

$$p_{t\,|\,0,1}\left(\mathbf{x} \mid \mathbf{x}_0, \mathbf{x}_1\right) \;=\; (1 - t)\,\delta_{\mathbf{x}_0}(\mathbf{x}) \;+\; t\,\delta_{\mathbf{x}_1}(\mathbf{x}), \tag{4}$$

where $\delta_{\mathbf{x}_0}(\mathbf{x})$ and $\delta_{\mathbf{x}_1}(\mathbf{x})$ are Dirac delta functions (analogous to one hot encodings) in the discrete space that allocate all probability mass to the fully masked and fully unmasked sequences at $t = 0$ and $t = 1$, respectively. For any intermediate value $t \in (0, 1)$, the interpolation governed by the weights $(1 - t)$ and $t$ yields a new video sequence $\mathbf{x}_t$ that represents a partially corrupted sequence. This is achieved by sampling each token from a mixture distribution where $1 - t$ represents the probability of a token being masked.

**Kolmogorov Equation in Discrete State Spaces.** In continuous-state models, one leverages the Continuity Equation (Song et al., 2021) to ensure that a vector field $u(\mathbf{x}_t, t)$ induces the desired probability transition between $p(\mathbf{x}_0)$ and $p(\mathbf{x}_1)$. The discrete counterpart is given by the Kolmogorov Equation (Campbell et al., 2024), which similarly characterizes how a probability distribution evolves in time over discrete states. To achieve a transition between the fully masked and fully unmasked video distributions, we define the vector field:

$$u_t(\mathbf{x}_t) \;=\; \frac{t}{1 - t}\left[p_{1\,|\,t}(\mathbf{x}_1 \mid \mathbf{x}_t, t; \theta) \;-\; \delta_{\mathbf{x}_t}(x)\right], \tag{5}$$

where $p_{1|t}(\mathbf{x}_1 \mid \mathbf{x}_t, t; \theta)$ is the model-predicted distribution of clean tokens $\mathbf{x}_1$ given a partially corrupted sequence $\mathbf{x}_t$ at time $t$. Here, $\delta_{\mathbf{x}_t}(x)$ represents the discrete Dirac delta centered at $\mathbf{x}_t$. By following $u_t$ through time, we recover a path that transforms $p(\mathbf{x}_0)$ into $p(\mathbf{x}_1)$.

## 3.3 Training with Frame-Level Masking

The flow matching formulation introduced in Sections 3.1 and 3.2 employs a single scalar timestep $t$ to interpolate between the fully masked and fully unmasked video distributions. Our training procedure uses a reparametrization of this timestep. In our method, videos are generated in chunks, and only a subset of the frames (the non-context frames) are sampled from a fully masked initial state. To better simulate this process during training, we reparametrize the global timestep $t$ into a per-frame timestep vector $\mathbf{t} = (t^1, \ldots, t^k)$ where each timestep $t^f$ specifies the masking ratio applied to frame $f$. In our setup, the context frames are assigned $t^f = 1$ (i.e. fully unmasked) while the new frames receive a masking level sampled from $\mathcal{U}(0, 1)$. By training the model to unmask frames with varying masking ratios per frame, we ensure that the network can effectively handle unmasked context frames while still learning a continuous transition from $p(\mathbf{x}_0)$ to $p(\mathbf{x}_1)$. To emphasize the reconstruction of masked tokens, we apply a masking operation on the cross-entropy loss. This results in the following objective:

$$\mathcal{L}_\theta = \mathbb{E}_{p(\mathbf{x}_1)\, p(\mathbf{x}_0)\, \mathcal{U}(\mathbf{t};0,1)\, p_{t|0,1}(\mathbf{x}_t \mid \mathbf{x}_0, \mathbf{x}_1)} \Big[ \underbrace{\delta_{[M]}(\mathbf{x}_t)\, (\mathbf{x}_1)^\top}_{\text{Loss Masking}}\, \log p_{1|t}(\mathbf{x}_1 \mid \mathbf{x}_t, \mathbf{t}; \theta) \Big], \qquad (6)$$

where $\delta_{[M]}(\mathbf{x}_t)$ indicates that only masked tokens are used in the cross-entropy computation. The choice of frame-level masking is essential because it aligns the task of generating chunks of size $k$ conditioned on $m$ clean context frames with our training task. In both scenarios, our models are tasked with unmasking frame sequences with varying masking levels across frames. We show that compared to a constant masking level baseline, this training choice enables autoregressive rollout to long sequence lengths. Our training algorithm is shown in the Appendix.

## 3.4 Chunkwise Autoregression for Long Videos

To generate a video of length $L \gg k$, we employ the chunkwise autoregressive approach previously described. Let $m$ be the number of context frames provided to the model (drawn initially from ground-truth, later from previous generated frames). In each iteration, we pass $k$ frames to the model, where the first $m$ frames are context and the remaining $(k-m)$ frames are fully masked. The model unmasks these frames. We then shift the context window forward by $s$ and repeat this process until we have generated $L$ total frames. Figure 2 illustrates this. Note that we dynamically increase the number of context frames $m$ in the final chunk in case there are less than $s$ frames left to generate. In those cases, we set $m = k - R$ where $R$ is the remaining number of frames, giving the final chunk a larger context. We do this to avoid generating video lengths beyond $L$ which would result in either discarding generated frames or generating videos longer than $L$. This is shown in Algorithm 1.

**Autoregressive *v.s.* Full-Sequence Generation.** By varying the stride $s$, we can interpolate between (i) a fully autoregressive mode ($s = 1$) with $m = k - s$, where we generate a single new frame per chunk, and (ii) a full-sequence mode ($s = k - m$), where we generate $k - m$ new frames simultaneously in each chunk. Smaller $s$ increases compute cost but may yield higher frame quality, whereas larger $s$ is more efficient, but may result in a drop in frame quality. Our experimental results shown in Table 2a support this intuition.

**FM-Style *v.s.* MGM-Style Sampling.** MaskFlow supports two distinct sampling modes. In FM-style sampling, we gradually traverse the probability path from the fully masked sequence $\mathbf{x}_0$ to the final unmasked sequence $\mathbf{x}_1$. A smaller step size yields smoother transitions at the cost of more denoising steps. Alternatively, in MGM-style sampling, we apply confidence-based heuristic sampling similar to Chang et al. (2022). In each sampling step, the model computes token-wise confidence scores for each predicted token and selects a fraction of the most confident tokens to unmask. This sampling process allows us to generate video chunks efficiently in fewer sampling steps.

**Timestep-dependent models and timestep-independent sampling.** By default, our model back-bones are timstep-dependent, meaning each forward pass receives a timestep vector $\mathbf{t} \in [0,1]^k$ that indicates the masking ratio of each frame. We embed $\mathbf{t}$ using a sinusoidal encoding followed by a learned MLP to produce conditioning vectors that modulate each transformer block via Adaptive Layer Normalization (Xu et al., 2019). Interestingly, we can still sample these models timstep-independently. Concretely, when using MGM-style sampling, we simply pass $\mathbf{t} = \mathbf{0}$ at each unmasking iteration, effectively treating our model as if it were timestep-independent:

$$p(\mathbf{x}_1|\mathbf{x}_t;\theta) \approx p(\mathbf{x}_1|\mathbf{x}_t, \mathbf{t} = \mathbf{0};\theta). \tag{7}$$

This works, since the learned network can infer the corruption state (mask ratio) from the input tokens alone. Thus, *a single* trained model can serve as a standard time-dependent (flow-matching) generator *and* as a time-independent (MGM-style) sampler, providing greater flexibility at inference time.

---

**Algorithm 1** Chunkwise Autoregression for Long Videos

---

**Require:** Video length $L$, context frames $\mathbf{x}^{1:m} = (x^1, \ldots, x^m)$,
1: chunk size $k$, stride $s$, fully masked frame $[M]$, network $p(\mathbf{x}_1 \mid \mathbf{x}_t, \mathbf{t}; \theta)$
2: **Initialize:** $\hat{\mathbf{x}}_1 \leftarrow (x^1, \ldots, x^m)$; $c \leftarrow m$                       ▷ current frame
3: **while** $c < L$ **do**
4:     $R \leftarrow L - c$                                  ▷ remaining frames
5:     $h \leftarrow \min(R, s)$                        ▷ frames to generate this chunk
6:     **if** $R \leq s$ **then**
7:         $m \leftarrow k - R$
8:     **end if**
9:     $\mathbf{x}_{\text{context}} \leftarrow (x^{c-m+1}, \ldots, x^c)$
10:    $\mathbf{x}_{\text{mask}} \leftarrow (\underbrace{[M], \ldots, [M]}_{h \text{ times}})$
11:    $\mathbf{x}_{\text{out}} \sim p\Big(\mathbf{x}_1 \mid (\mathbf{x}_{\text{context}}, \mathbf{x}_{\text{mask}}), \mathbf{t}; \theta\Big)$
12:    $\mathbf{x}_{\text{new}} \leftarrow (x_{\text{out}}^{m+1}, \ldots, x_{\text{out}}^{m+h})$
13:    $\hat{\mathbf{x}}_1 \leftarrow (\hat{\mathbf{x}}_1, \mathbf{x}_{\text{new}})$
14:    $c \leftarrow c + h$
15: **end whilereturn** $\hat{\mathbf{x}}_1$

---

# 4 Experiments

## 4.1 Datasets and Evaluation Metrics

**Datasets.** We evaluate MaskFlow on two datasets: Deepmind Lab (DMLab) for evaluating performance in diverse ego-centric views and FaceForensics (FFS) for assessing video fluency. DMLab contains videos of random walks in a 3D maze, while FFS consists of deepfakes. Both datasets are preprocessed and tokenized using SD-VQGAN (Rombach et al., 2022) for training. Due to computational constraints, we focus on low-resolution datasets, which allow us to thoroughly evaluate long-horizon sampling behavior while keeping training and inference tractable. Our approach is resolution-agnostic and can be applied to higher-resolution datasets with appropriate resources. Further details are provided in the Appendix.

**Evaluation metrics: FVD for video quality, NFE for sampling efficiency.** For video generation, we use Fréchet Video Distance (FVD) (Unterthiner et al., 2018) as our main evaluation metric. For FVD, we adhere to the evaluation guidelines introduced by StyleGAN-V (Ma et al., 2024; Skorokhodov et al., 2022). We additionally evaluate the sampling efficiency of our method against various baselines by comparing the required number of function evaluations (NFE) and sampling wall clock times using identical compute resources.

**Baselines.** The two most comparable works to our method are Chen et al. (2024) and Ruhe et al. (2024). Both of these techniques propose novel sampling methods that can be rolled out to long video lengths, and also apply frame-specific noise levels. Both of these approaches are diffusion-based and operate on continuous representations, whereas we operate on discrete tokens and use masking. Although there are many other discrete video generation works like MAGViT (Yu et al., 2023a) or Muse (Chang et al., 2023), our method focuses on long video generation specifically and primarily shines when generating longer sequences. We re-implement both the pyramid sampling scheme proposed in Diffusion Forcing and the Rolling Diffusion sampling method in our discrete setting. This allows us to compare the baseline sampling methods with MaskFlow on the same model backbones. We also compare MaskFlow with a constant masking level baseline to evaluate the design choice of frame-level masking.

## 4.2 Main Results

| Sampling Mode | Extrapolation Factor | Total NFE | FVD ↓ |
|---|---|---|---|
| Diffusion Forcing Chen et al. (2024) | 2× | 798 | 144.43 |
| Rolling Diffusion Ruhe et al. (2024) | 2× | 750 | 72.49 |
| *MaskFlow* (FM-Style) | 2× | 788 | 66.94 |
| *MaskFlow* (MGM-Style) | 2× | **60** | **59.93** |
| Diffusion Forcing Chen et al. (2024) | 5× | 1,596 | 272.14 |
| Rolling Diffusion Ruhe et al. (2024) | 5× | 1,652 | 248.13 |
| *MaskFlow* (FM-Style) | 5× | 1,500 | 118.81 |
| *MaskFlow* (MGM-Style) | 5× | **120** | **108.74** |
| Diffusion Forcing Chen et al. (2024) | 10× | 3,192 | 306.31 |
| Rolling Diffusion Ruhe et al. (2024) | 10× | 3,092 | 451.38 |
| *MaskFlow* (FM-Style) | 10× | 3,000 | **174.85** |
| *MaskFlow* (MGM-Style) | 10× | **240** | 214.39 |

Table 1: **Both MGM-style and FM-style sampling extrapolate to longer sequences with similar FVD, but MGM-style is much faster.** MaskFlow consistently outperforms Diffusion Forcing and Rolling Diffusion. Results are on FFS with $s = k - m$.

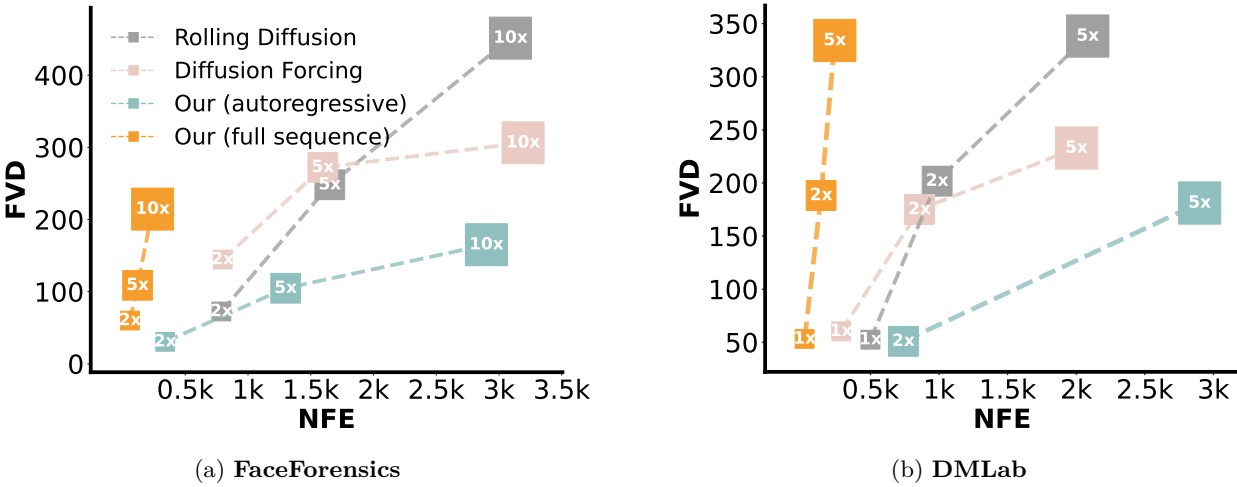

(a) **FaceForensics**

(b) **DMLab**

Figure 3: **MaskFlow performance scales favorably across NFE for different extrapolation factors across both datasets.** A comparison between MaskFlow (full sequence & autoregressive) and other baselines at different extrapolation factors.

**Our MGM-style sampling approach can generate long videos efficiently with minimal degradation.** Table 1 shows the ability of our model to generate long videos. We define *extrapolation factor* as the ratio of sampling and training window lengths, so an extrapolation factor of $2\times$ means we generate videos twice as long as the training videos. The experiments in the Appendix all use full sequence generation with $s = k - m$. Although video quality deteriorates for longer extrapolation factors due to error accumulation, our method is able to maintain visual quality on par with or higher than baselines for large extrapolation factors at much lower NFE. This ability is specifically enabled by our frame-level masking approach, which ensures that our models are able to unmask arbitrary mixtures of low- and high-masking ratio frames. This allows us to condition each chunk on arbitrary numbers of previously generated frames, which is consistent with the training task. A detailed qualitative overview is shown in Figure 5. Both FM-style and MGM-style sampling modes retain this ability, but our MGM-style sampling generates high-quality results with significantly lower NFE. We also show that MaskFlow outperforms both Rolling Diffusion (Ruhe et al., 2024) and Diffusion Forcing (Chen et al., 2024) in discrete settings.

**Frame-level masking does not reduce performance on original training window length generation.** Table 3 shows that our frame-level masking approach does not reduce performance for a single chunk compared to a constant masking baseline. We compare a frame-level masking DMLab model trained on $k = 36$ frames with a constant masking baseline and show that our frame-level masking models outperform the constant masking baseline across two sampling modes. This demonstrates that our frame-level masking training does not trade off quality on training window length generation for the ability to generate longer videos.

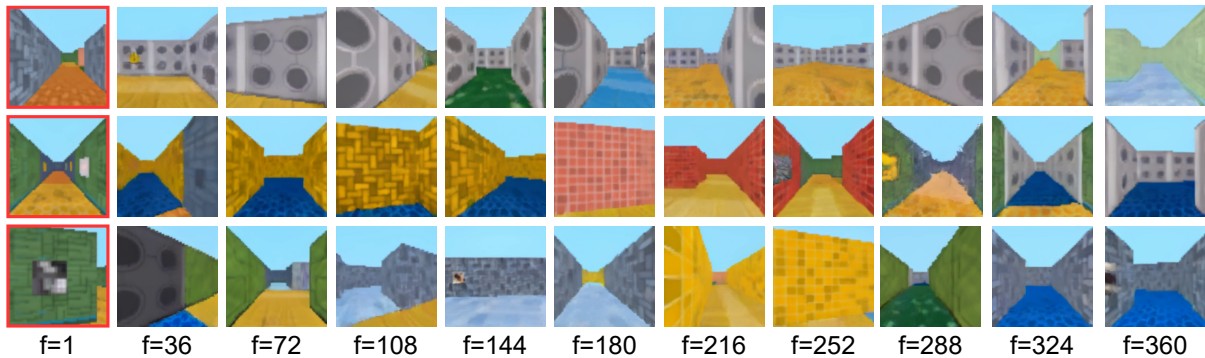

f=1    f=36    f=72    f=108    f=144    f=180    f=216    f=252    f=288    f=324    f=360

Figure 4: **Autoregressive sampling stabilizes DMLab videos beyond extrap. factor** $10\times$**.** All examples use autoregressive MaskFlow (MGM-style) sampling. Final context frame is shown in red.

**Fully autoregressive sampling increases quality but sacrifices speed.** To further illustrate the flexibility of our method, we run a series of experiments using a sampling stride of $s = 1$ with $m = k - 1$. We thus initialize the generative process by conditioning on almost a full training clip, and then generating new frames frame by frame using our existing sampling approaches. This requires us to traverse the entire unmasking chain for each generated frame, making this sampling method slower than the sampling approach employed in Table 1. Specifically on DMLab, which is more dynamic than FFS, this substantially improves results, enabling extremely long high-quality rollouts (see Figure 4. The findings in Table 2a show that autoregressive generation yields consistent improvements across datasets at the cost of efficiency. However, since our MGM-style sampling is able to generate new frames in very few NFE, autoregressive frame-by-frame generation in MaskFlow actually requires a similar NFE than the baselines that do full sequence generation with FM-style sampling. Figure 3 highlights this, showing that MaskFlow scales favorably compared to other methods in terms of NFE for $s = 1$ and $s = k - m$. A more detailed comparison of autoregressive and full sequence sampling in terms of wall clock sampling speed can be found in the Appendix. We hypothesize that this improvement in quality is because next frame unmasking is a less challenging task than full sequence unmasking, and that error accumulation is slower when models receive a larger number of context frames in each forward pass.

|  | Extrapolation Factor | Sampling Stride | Total NFE | FVD ↓ |
|---|---|---|---|---|
| FaceForensics | 2× | $s = 14$ (*full sequence*) | **60** | 59.93 |
| FaceForensics | 2× | $s = 1$ (*autoregressive*) | 340 | **30.43** |
| FaceForensics | 5× | $s = 14$ (*full sequence*) | **120** | 108.74 |
| FaceForensics | 5× | $s = 1$ (*autoregressive*) | 1,300 | **103.69** |
| FaceForensics | 10× | $s = 14$ (*full sequence*) | **240** | 214.39 |
| FaceForensics | 10× | $s = 1$ (*autoregressive*) | 2,900 | **165.02** |
| DMLab | 2× | $s = 24$ (*full sequence*) | **60** | 195.84 |
| DMLab | 2× | $s = 1$ (*autoregressive*) | 740 | **42.53** |
| DMLab | 5× | $s = 24$ (*full sequence*) | **140** | 334.15 |
| DMLab | 5× | $s = 1$ (*autoregressive*) | 2,900 | **80.56** |

(a) Fully autoregressive sampling vs. full-sequence sampling improves performance at cost of higher NFE.

| Extrapolation Factor | Guidance Level $\omega$ | FVD ↓ DMLab |
|---|---|---|
| 1× | 0 | **45.84** |
| 1× | 1.0 | 49.76 |
| 1× | 2.0 | 46.29 |
| 2× | 0 | 219.33 |
| 2× | 1.0 | 189.48 |
| 2× | 2.0 | **141.94** |
| 5× | 0 | 402.73 |
| 5× | 1.0 | 403.32 |
| 5× | 2.0 | **281.20** |

(b) Partial context guidance improves long-extrapolation results.

Table 2: **Comparison of sampling strategies and guidance levels across extrapolation factors on FaceForensics and DMLab.**

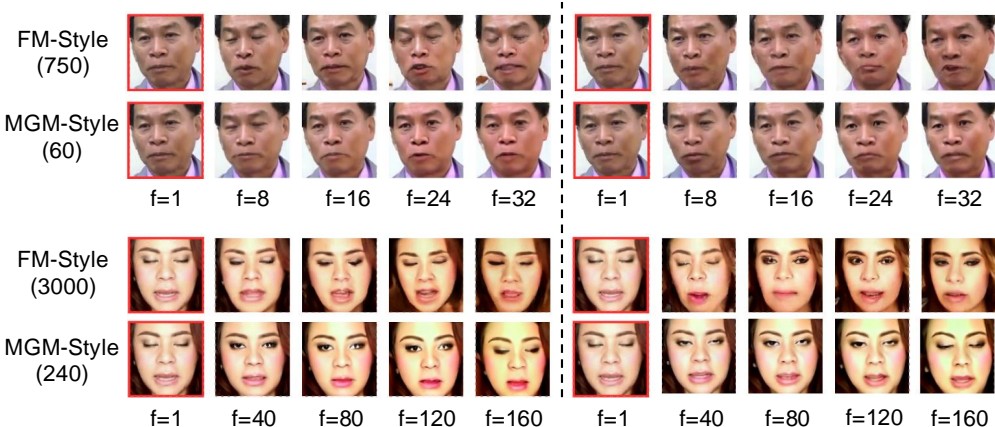

Figure 5: **MGM-style sampling generates visually pleasing videos with two context frames beyond** $10\times$ **training frame length with few sampling steps.** Shows sampling mode and total NFE in brackets, and frame indices $f$.

**Scaling partial context guidance further improves performance on full sequence generation.** Inspired by classifier-free guidance (Ho & Salimans, 2021) and history guidance in Diffusion Forcing (Chen et al., 2024; Song et al., 2025), we propose a training-free sampling method that fuses multiple model predictions of $p(x_1|x_t; \theta)$ using different levels of conditioning on past frames. Concretely, we run forward passes where $x_t$ contains: (i) *no* context frames (unconditional) , (ii) *partially masked* context frames (partial conditioning), and (iii) *fully clean* context frames (fully conditional). We then fuse the predicted logits with a guidance scale $\omega$. By using *partially masked* rather than fully clean context frames for some of these passes, the model is encouraged to preserve global movement and dynamics without strictly copying the observed context. Formally, if $z_{\text{uncond}}(i)$, $z_{\text{partial}}(ii)$, $z_{\text{cond}}(iii)$ denotes logits from the three forward passes, one can construct a composite logit distribution via $z_{\text{cond}} + \omega \cdot (z_{\text{partial}} - z_{\text{uncond}})$ that balances sample variety (unconditional) with temporal coherence (partial and full context). Partial context guidance requires no retraining and can yield improved fidelity and motion consistency. Table 2b shows performance improvements achieved on timestep-independent DMLab models.

### 4.3 Ablations

**MGM-style and FM-style NFE choices minimize visual quality and sampling efficiency trade-offs.** The choice of NFE in our work is driven empirically. We compare generation quality when generating a single chunk $k$ on both datasets and tune our NFE accordingly for FM-style and MGM-style sampling

| Training Mode | Sampling Mode (NFE) | FVD ↓ DMLab |
|---|---|---|
| Constant Masking | FM-Style | 53.31 |
| Frame-level Masking | Diffusion Forcing (Chen et al., 2024) | 60.30 |
| Frame-level Masking | Rolling Diffusion (Ruhe et al., 2024) | 52.43 |
| Frame-level Masking | *MaskFlow* (Ours) | **49.62** |

Table 3: **Frame-level masking outperforms constant masking on training window length**.

modes. We are aware that our observations regarding sampling speeds depend on the choice of NFE, so we compare video quality for a lower number of sampling steps for both sampling modes on both datasets. In Figure 6, we show that our choices of 20 for MGM-style and 250 for FM-style sampling achieve the best trade-off between sampling efficiency and quality, since video quality saturates for higher NFE in both modes across both datasets.

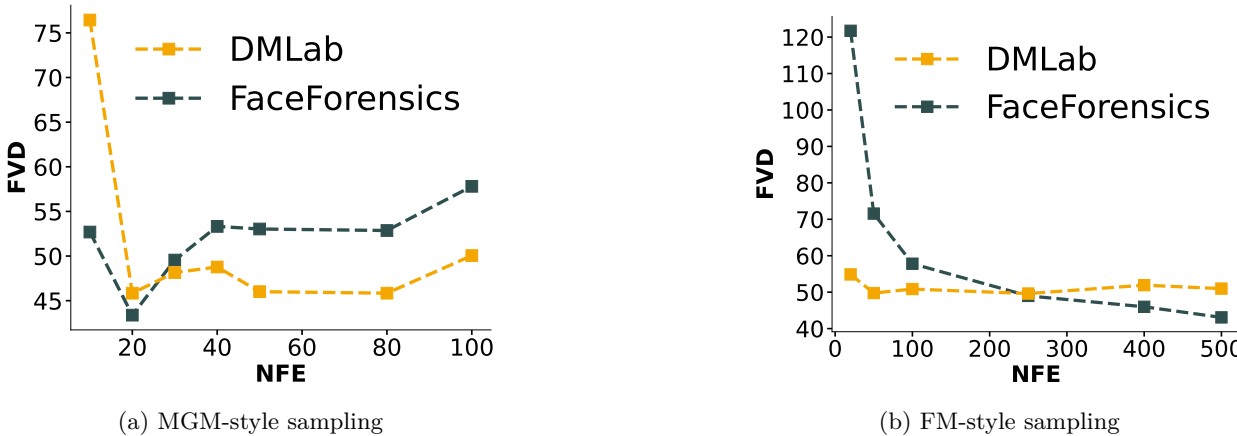

(a) MGM-style sampling

(b) FM-style sampling

Figure 6: **NFE choices for both MGM-style (20) and FM-style (250) suitably trade off sampling speed with visual quality.** Figures show FVD on a single chunk for frame-level masking models.

**Timestep-dependent models can be sampled in a time-independent training-free manner.** An additional interesting observation is that MGM-style sampling without explicit timestep conditioning is able to generate high-quality results in the full-sequence case. We thus compare timestep-dependent and timestep-independent models under different sampling modes in Table S8. Our results demonstrate that the timestep-depe ndent models when sampled with MGM-style sampling actually perform best. We hypothesize that this is due to the more explicit inductive bias of timestep conditioning during training, and that this guides the learning process towards improved unmasking irrespective of the actual timesteps passed during inference. We are thus able to apply our sampling modes across timestep-dependent and independent models without requiring any re-training.

## 5 Conclusion and Limitations

We have presented a discrete flow matching framework for flexible long video generation that leverages frame-level masking during training. Our experiments demonstrate that this approach can generate high-quality videos beyond 10× the training window length, while substantially reducing sampling cost through MGM-style unmasking. Despite this, MaskFlow has some clear limitations. Firstly, error accumulation in autoregressive sampling modes remains a substantial challenge that is not solved in this approach. Secondly, we have used relatively low resolution datasets in our evaluations. Although there is nothing inherent to MaskFlow that limits its applicability to low-resolution datasets, the method's performance in higher

| Sampling Mode | Model Time Dep. | Sampling-Time Indep. | Extrap. Factor | FVD ↓ | |
|---|---|---|---|---|---|
| | | | | DMLab | FaceForensics |
| FM-Style | ✓ | ✗ | 1× | 55.19 | 48.98 |
| MGM-Style | ✗ | ✓ | 1× | **45.84** | 77.04 |
| MGM-Style | ✓ | ✓ | 1× | 53.17 | **45.92** |
| FM-Style | ✓ | ✗ | 2× | 267.80 | 66.94 |
| MGM-Style | ✗ | ✓ | 2× | 219.33 | 109.96 |
| MGM-Style | ✓ | ✓ | 2× | **188.22** | **59.93** |
| FM-Style | ✓ | ✗ | 5× | 360.61 | 118.81 |
| MGM-Style | ✗ | ✓ | 5× | 402.73 | 137.66 |
| MGM-Style | ✓ | ✓ | 5× | **334.15** | **108.74** |

Table 4: **Timestep-dependent models can generate high-quality results with timestep-independent sampling.** Timestep-dependent models with timestep-independent sampling show best results across various extrapolation factors.

resolutions and more dynamic real-world datasets is not fully validated. Despite this, we believe that discrete tokens have a lot of potential for high-quality, efficient visual generation.

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

# A    Appendix of MASKFLOW: Discrete Flow For Flexible and Long Video Generation

## A.1    Additional Related Work

**Masked Diffusion Models.**   Limitations of autoregressive models for probabilistic language modeling have recently sparked increasing interest in masked diffusion models. Recent works like Shi et al. (2024) and Sahoo et al. (2024) have aligned masked generative models with the design space of diffusion models by formulating continuous-time forward and sampling processes. Works like Nie et al. (2024) and Gong et al. (2024) also demonstrate the significant scaling potential of MDM for language tasks, indicating that this masked modeling paradigm can rival autoregressive approaches for modalities beyond language such as protein co-design Campbell et al. (2024) and vision.

## A.2    Computation of NFE for Different Sampling Methods

Our sampling speed evaluations are determined by computing the required number of chunks

$$\ell = \left\lceil \frac{L - k}{s} \right\rceil + 1,$$

to generate a video of total length $L$, where $k$ is the chunk size and $s$ is the stride with which the chunk start is shifted. The overall number of function evaluations (NFEs) is then obtained by multiplying $\ell$ with the number of sampling steps required to generate one chunk. We apply this methodology for all chunkwise-autoregressive approaches.

- **MGM-Style Sampling:** In this method each chunk is generated in 20 forward passes, so that the total NFE is
$$\text{NFE}_{\text{MGM}} = \ell \times 20.$$

- **FM-Style Sampling:** Here we generate each chunk in 250 forward passes:
$$\text{NFE}_{\text{FM}} = \ell \times 250.$$

- **Diffusion Forcing with Pyramid Scheduling:** Here, we apply 250 sampling timesteps per frame but begin unmasking earlier frames as the denoising process proceeds. For a chunk of $k$ frames, we generate a scheduling matrix with
$$H = 250 + (k - 1) + 1 = k + 250$$

rows and $k$ columns. Each entry in the scheduling matrix is computed as

$$\text{scheduling\_matrix}[i,j] = 250 + j - i, \quad \text{for } i = 0, \ldots, H-1 \text{ and } j = 0, \ldots, k-1,$$

and then clipped to the interval $[0, 249]$. Since we iterate through each of the $H$ rows of the denoising matrix in each chunk we effectively compute

$$\text{NFE}_{\text{DiffusionForcing}} = k + 250.$$

- **RDM Sampling:** This approach proceeds in three stages:

  1. *Initialization (Init-Schedule):* The initial window of $k$ frames is processed using a fixed schedule that applies $T = 250$ forward passes to bring the window to its rolling state.
  2. *Sliding Window Handling:* After initialization, the window is shifted by one frame at a time. For each shift, an inner loop is executed that updates the denoising levels until the first non-context frame (i.e., the frame immediately following the $m$ context frames) is fully denoised (i.e., reaches a value of 1). This inner loop requires $\left\lceil \frac{T}{k-m} \right\rceil$ forward passes per window shift. As the window is shifted $(L-k)$ times, this stage contributes roughly $(L-k) \times \left\lceil \frac{T}{k-m} \right\rceil$ forward passes.
  3. *Final Window Processing:* Once the sliding window stage is complete, the final (partial) window is further refined until all frames are fully denoised. This final stage requires additional 250 forward passes.

  Thus, the total NFE for RDM is given by

  $$\text{NFE}_{\text{Rolling}} = 250 \text{ (init-schedule)} + (L-k) \times \left\lceil \frac{T}{k-m} \right\rceil \text{ (sliding)} + 250 \text{ (final window)}.$$

## A.3   Training & Implementation Details

We use a vocabulary size $K = 16{,}384$ and token length 1,024 to compress video frames by a compression factor of 8. We then train on a small subset of training sequences of $k = 16$ frames for FFS and $k = 36$ frames for DMLab. We use a Latte XL2 Ma et al. (2024) backbone with 760M parameters for all FFS experiments, and a smaller Latte B2 backbone architecture with 129M parameters for DMLab, and train it using discrete flow matching dynamics.

All FFS models were trained on 4 H100 GPUs with a local batch size of 4. We run training for a total of 200,000 steps and use a sigmoid scheduler that determines the per-frame masking ratio for a sampled masking level $t^k$. We use an AdamW optimizer with a learning rate of $1e-4$ and $\beta_1 = 0.9$ and $\beta_2 = 0.999$. We additionally incorporate a frame-level loss weighting mechanism based that is also based on $t^k$. We adopt *fused*-SNR loss weighting from Hang et al. (2023); Chen et al. (2024) and derive it for discrete flow matching. Let

$$\text{SNR}(t) \;=\; \frac{\kappa(t)^2}{1 - \kappa(t)^2},$$

where $\kappa(t)$ is the masking schedule. The *fused*-SNR mechanism smoothes SNR values across time steps in a video by computing an exponentially decaying SNR from previous frames (or tokens). We refer the reader to for full details.

## A.4   Baseline Details

The two most comparable works to our method are Chen et al. (2024) and Ruhe et al. (2024). Both of these techniques propose novel sampling methods that can be rolled out to long video lengths, and also apply frame-specific noise levels. Both of these approaches are diffusion-based and operate on continuous

| Sampling Mode | Stride | Extrap. Factor | Total NFE | Sampling Time [s] | FVD DMLab↓ | FVD FFS↓ |
|---|---|---|---|---|---|---|
| Diffusion Forcing | $s = k - m$ | 2× | 858/798 | 135.97/156.78 | 175.01 | 144.43 |
| Rolling Diffusion | $s = k - m$ | 2× | 896/788 | 141.99/154.81 | 201.70 | 72.49 |
| *MaskFlow* (MGM-Style) | $s = k - m$ | 2× | **60/60** | **9.51/9.30** | 188.02 | 59.93 |
| *MaskFlow* (MGM-Style) | $s = 1$ | 2× | 740/340 | 117.27/66.80 | **50.87** | **30.43** |
| Diffusion Forcing | $s = k - m$ | 5× | 2,002/1,596 | 317.27/313.56 | 232.89 | 272.14 |
| Rolling Diffusion | $s = k - m$ | 5× | 2,084/1,652 | 330.27/324.56 | 338.34 | 248.13 |
| *MaskFlow* (MGM-Style) | $s = k - m$ | 5× | **140/120** | **22.19/23.58** | 334.15 | 108.74 |
| *MaskFlow* (MGM-Style) | $s = 1$ | 5× | 2,900/1,300 | 100.09/379.91 | **181.11** | **103.69** |

Table S5: **MGM-Style sampling is much faster without sacrificing quality.** We report total NFE, sampling time (s), and FVD on DMLab and FaceForensics (FFS) across various sampling modes and extrapolation factors.

---

**Algorithm 2 FM-Style Sampling with Context Frames for a Single Chunk**

---

**Require:** $p(\mathbf{x}_1|\mathbf{x}_t, \mathbf{t}; \theta)$, $t$, context frames $\mathbf{c} = (c^1, \ldots, c^m)$, fully masked frame $[M]$ (i.e., a frame where every token equals the mask token $M$), $t \in [0, 1]$, $\Delta t$

1: $\mathbf{x}_t \leftarrow (c^1, \ldots, c^m, [M], \ldots, [M])$
2: $t \leftarrow 0$
3: $\mathbf{t} \leftarrow (1, \ldots, 1, 0, \ldots 0)$
4: **while** $t \leq 1 - \Delta t$ **do**
5: $\quad u_t(\mathbf{x}_t) = \frac{t}{1-t}\Big[p_\theta(\mathbf{x}_1 \mid \mathbf{x}_t, \mathbf{t}) - \delta_{\mathbf{x}_t}\Big]$
6: $\quad p_\theta(\mathbf{x}_1 \mid \mathbf{x}_{t+\Delta t}, \mathbf{t} + \Delta t) = \text{Cat}\Big[\delta_{\mathbf{x}_t} + u_t(\mathbf{x}_t)\,\Delta t\Big]$
7: $\quad$ **For each token** $n$ in $\mathbf{x}_t$:
8: $\quad x_{t+\Delta t}^n \leftarrow \begin{cases} x_t^n, & \text{if } x_t^n \neq M, \\ p(\cdot|\mathbf{x}_{t+\Delta t}, \mathbf{t} + \Delta t; \theta), & \text{if } x_t^n = M. \end{cases}$
9: $\quad t \leftarrow t + \Delta t$
10: $\quad \mathbf{t} \leftarrow \mathbf{t} + \Delta t$
11: **end whilereturn** $\mathbf{x}_t$

---

representations, whereas we operate on discrete tokens and use masking. We re-implement both the pyramid sampling scheme proposed in Diffusion Forcing and the Rolling Diffusion sampling method in our discrete setting. This allows us to compare the baseline sampling methods to MaskFlow on the same model backbones. To isolate the effect of our chunkwise autoregressive sampling methodology on performance from the effects of tokenization, we reimplement both the pyramid sampling scheme proposed in Diffusion Forcing and the Rolling Diffusion sampling method for our discrete setting. This allows us to compare the baseline sampling methods on the same timestep-dependent model backbone. Although it is conceivable that Rolling Diffusion sampling may perform better when applied to a model explicitly trained using the progressive noise schedule suggested in Ruhe et al. (2024), we believe this comparison is still fair. Our training methodology does not inject any inductive bias by way of the masking level into the model, so there is no obvious advantage that our sampling should have over other methods. We provide a comprehensive evaluation of performance and sampling efficiency across both datasets and different sampling modes.

## A.5 Dataset Details

**Deepmind Lab.** The Deepmind Lab (DMLab) navigation dataset contains $64 \times 64$ resolution videos of random walks in a 3D maze environment. We use the total 625 videos with frame length 300 frames, and randomly sample sequences of 36 consecutive frames from each video during training. We upscale video frames to a resolution of $256 \times 256$ before tokenizing them similar to our approach for FaceForensics. We disregard the provided actions, focusing on action-unconditional video generation. We use $m = 12$ and $s = 24$ for the DMLab full sequence generation experiments unless stated otherwise.

---

**Algorithm 3 MGM-Style Sampling for a Single Chunk**

---

**Require:** Network $p(\mathbf{x}_1 \mid \mathbf{x}_t, \mathbf{t}; \theta)$, context frames $\mathbf{c} = (c^1, \ldots, c^m)$, masked frame $[M]$ (i.e., every token equals $M$), total unmasking steps $T$

1: **Initialize:**
2: $\mathbf{x}_t \leftarrow (\mathbf{c}, [M], \ldots, [M])$
3: $\mathbf{t} \leftarrow (\underbrace{1, \ldots, 1}_{m}, \underbrace{0, \ldots, 0}_{k-m})$
4: Define the set of masked token indices in $\mathbf{x}_t$:
5: $\mathcal{M} \triangleq \{ n \mid x_t^n = M \}$.
6: **for** $i = 1$ **to** $T$ **do**
7:     Compute token-wise logits:
8:     $\boldsymbol{\lambda} \leftarrow p(\mathbf{x}_1 \mid \mathbf{x}_t, \mathbf{t}; \theta)$.
9:     **For each token** $n \in \mathcal{M}$:
10: sample $\hat{x}_t^n \sim \mathrm{Cat}\Big(\mathrm{Softmax}(\boldsymbol{\lambda}^n)\Big)$
11: and compute the confidence score $C_n = \mathrm{Softmax}(\boldsymbol{\lambda}^n)_{\hat{x}_t^n}$.
12:
13:     **Define the confidence threshold:**
14: Let $\alpha$ denote the desired fraction of masked tokens to update in each iteration (e.g. $\alpha = 1/T$).
15: Then set $\tau_c = \min\Big\{ c \in [0,1] \ \Big| \ \big|\{j \in \mathcal{M} \mid C_j \geq c\}\big| \geq \lceil \alpha |\mathcal{M}| \rceil \Big\}$.
16: (That is, $\tau_c$ is chosen as the minimum confidence such that at least $\lceil \alpha |\mathcal{M}| \rceil$ tokens have confidence scores at or above $\tau_c$, thereby selecting the top $\lceil \alpha |\mathcal{M}| \rceil$ tokens.)
17:     **For each token** $n \in \mathcal{M}$ with $C_n \geq \tau_c$, update:
18: $x_t^n \leftarrow \hat{x}_t^n$.
19:     Update the set of masked indices:
20: $\mathcal{M} \leftarrow \{ n \mid x_t^n = M \}$.
21:     **if** $\mathcal{M} = \varnothing$ **then**
22:         **break**
23:     **end if**
24: **end for** **return** $\mathbf{x}_t$.

---

**FaceForensics.** FaceForensics (FFS) is a dataset that contains $150 \times 150$ images of deepfake faces, totaling 704 videos with varying number of frames at 8 frames-per-second. We upsample the resolution to $256 \times 256$, before encoding individual frames using the image-based tokenizer SD-VQGAN Rombach et al. (2022). While image-based tokenizers have shown to lead to flickering issues, we observe high-reconstruction quality (reconstruction FVD $\approx 8$ on FFS) on our datasets and thus leave work on video tokenization to other works. After tokenization, we train on encoded frame sequences of 16 frames, each consisting of token grids with dimensionality $32 \times 32$. We generally use $m = 2$ ground-truth context frames for conditioning, and $s = 14$.

### A.6 Training with Frame-level Masking

---

**Algorithm 4 Training with Frame-level Masking**

---

**Require:** Dataset of tokenized video clips $\mathcal{D}$, network $p(\mathbf{x}_1 \mid \mathbf{x}_t, \mathbf{t}; \theta)$, chunk size $k$

1: **while** not converged **do**
2:     **Sample** a chunk of $k$ frames from $\mathcal{D}$, denoted $\mathbf{x}_1 = (x_1^1, x_1^2, \ldots, x_1^k)$
3:     **for** $f = 1, \ldots, k$ **do**
4:         $t_f \sim \mathcal{U}(0,1)$
5:         $x_{t^f} \sim p_{t^f|0,1}\big( \cdot \mid x_0^f, x_1^f \big)$, where $p_{t^f|0,1}$ follows $(1 - t^f)\,\delta_{x_0^f} + t^f\,\delta_{x_1^f}$.
6:     **end for**
7:     $\mathbf{x}_t = (x_{t^1}^1, x_{t^2}^2, \ldots, x_{t^k}^k)$
8:     $\hat{\mathbf{x}}_1 = p\big(\mathbf{x}_1 \mid \mathbf{x}_t, \mathbf{t}; \theta\big)$, where $\mathbf{t} = (t^1, \ldots, t^k)$
9:     **Backpropagate** $\mathcal{L}_\theta(\mathbf{x}_1, \hat{\mathbf{x}}_1)$ and **update** $\theta$.
10: **end while**

---

### A.7 Further Quantitative Results

**Our chunkwise autoregressive MGM-style sampling is preferable to full sequence training in settings with limited hardware.** To evaluate our method for long video generation against a longer training window baseline, we compare the performance of a frame-level masking model trained on 16 frames with full sequence generation of a constant-masking level model trained on 32 frames with similar batch size and on similar hardware. In Table S6 we show that iterative rollout of our MGM-style sampling outperforms full sequence generation even when the full sequence model is trained on a longer window.

| Sampling Mode | Training Window | Sampling Window | Total NFE | FVD $\downarrow$ |
|---|---|---|---|---|
| FM-Style (bs=2) | 32 | 32 | 250 | 253.08 |
| *MaskFlow* (MGM-Style) (bs=2) | 16 | 32 | 60 | 192.76 |
| *MaskFlow* (MGM-Style) (bs=4) | 16 | 32 | 60 | **59.93** |

Table S6: **Our MGM-style sampling is more efficient and generates better results over baseline for larger training windows**. We train a constant masking ratio model on larger window sizes with similar batch size on similar hardware, and compare full sequence generation to generating the same length using our chunkwise MGM-style sampling.

| | Extrapolation Factor | Sampling Stride | Total NFE | FVD $\downarrow$ |
|---|---|---|---|---|
| FaceForensics | 2× | $s = 14$ (*full sequence*) | **60** | 59.93 |
| FaceForensics | 2× | $s = 1$ (*autoregressive*) | 340 | **30.43** |
| FaceForensics | 5× | $s = 14$ (*full sequence*) | **120** | 108.74 |
| FaceForensics | 5× | $s = 1$ (*autoregressive*) | 1,300 | **103.69** |
| FaceForensics | 10× | $s = 14$ (*full sequence*) | **240** | 214.39 |
| FaceForensics | 10× | $s = 1$ (*autoregressive*) | 2,900 | **165.02** |
| DMLab | 2× | $s = 24$ (*full sequence*) | **60** | 188.22 |
| DMLab | 2× | $s = 1$ (*autoregressive*) | 740 | **50.87** |
| DMLab | 5× | $s = 24$ (*full sequence*) | **140** | 334.15 |
| DMLab | 5× | $s = 1$ (*autoregressive*) | 2,900 | **181.11** |

Table S7: **Autoregressive sampling outperforms full sequence sampling on timestep-dependent models at the cost of higher NFE.**

| Sampling Mode | Model Time Dep. | Sampling-Time Indep. | Extrap. Factor | FVD ↓ | |
|---|---|---|---|---|---|
| | | | | DMLab | FaceForensics |
| FM-Style | ✓ | ✗ | 1× | 55.19 | 48.98 |
| MGM-Style | ✗ | ✓ | 1× | **45.84** | 77.04 |
| MGM-Style | ✓ | ✓ | 1× | 53.17 | **45.92** |
| FM-Style | ✓ | ✗ | 2× | 267.80 | 66.94 |
| MGM-Style | ✗ | ✓ | 2× | 219.33 | 109.96 |
| MGM-Style | ✓ | ✓ | 2× | **188.22** | **59.93** |
| FM-Style | ✓ | ✗ | 5× | 360.61 | 118.81 |
| MGM-Style | ✗ | ✓ | 5× | 402.73 | 137.66 |
| MGM-Style | ✓ | ✓ | 5× | **334.15** | **108.74** |

Table S8: **Timestep-dependent models can generate high-quality results with timestep-independent sampling.** Timestep-dependent models with timestep-independent sampling show best results across various extrapolation factors.

| | Extrapolation Factor | Sampling Stride | Total NFE | FVD ↓ |
|---|---|---|---|---|
| FaceForensics | 2× | $s = 14$ (*full sequence*) | **60** | 109.96 |
| FaceForensics | 2× | $s = 1$ (*autoregressive*) | 340 | **43.91** |
| FaceForensics | 5× | $s = 14$ (*full sequence*) | **120** | **137.66** |
| FaceForensics | 5× | $s = 1$ (*autoregressive*) | 1,300 | 193.90 |
| FaceForensics | 10× | $s = 14$ (*full sequence*) | **240** | **174.92** |
| FaceForensics | 10× | $s = 1$ (*autoregressive*) | 2,900 | 293.16 |
| DMLab | 2× | $s = 24$ (*full sequence*) | **60** | 219.33 |
| DMLab | 2× | $s = 1$ (*autoregressive*) | 740 | **42.53** |
| DMLab | 5× | $s = 24$ (*full sequence*) | **140** | 402.73 |
| DMLab | 5× | $s = 1$ (*autoregressive*) | 2,900 | **80.56** |

Table S9: **Autoregressive sampling outperforms full sequence sampling on timestep-independent models at the cost of higher NFE.** Performance improvement on DMLab is substantial.

## A.8 Further Qualitative Results

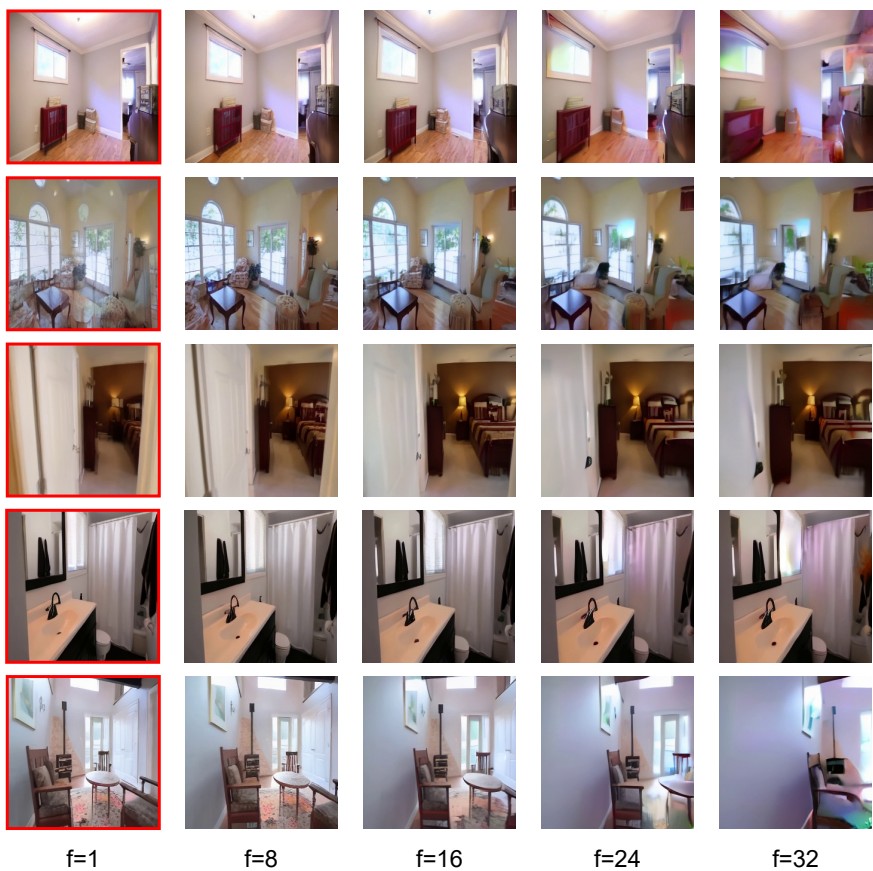

f=1  f=8  f=16  f=24  f=32

Figure S7: **Further visualizations on the Realestate10K Zhou et al. (2018) dataset.** Models trained on chunk size $k = 16$ with 4 H100 GPUs. Due to computational limitations, we cannot provide further analyses on this larger, more compute intensive dataset.

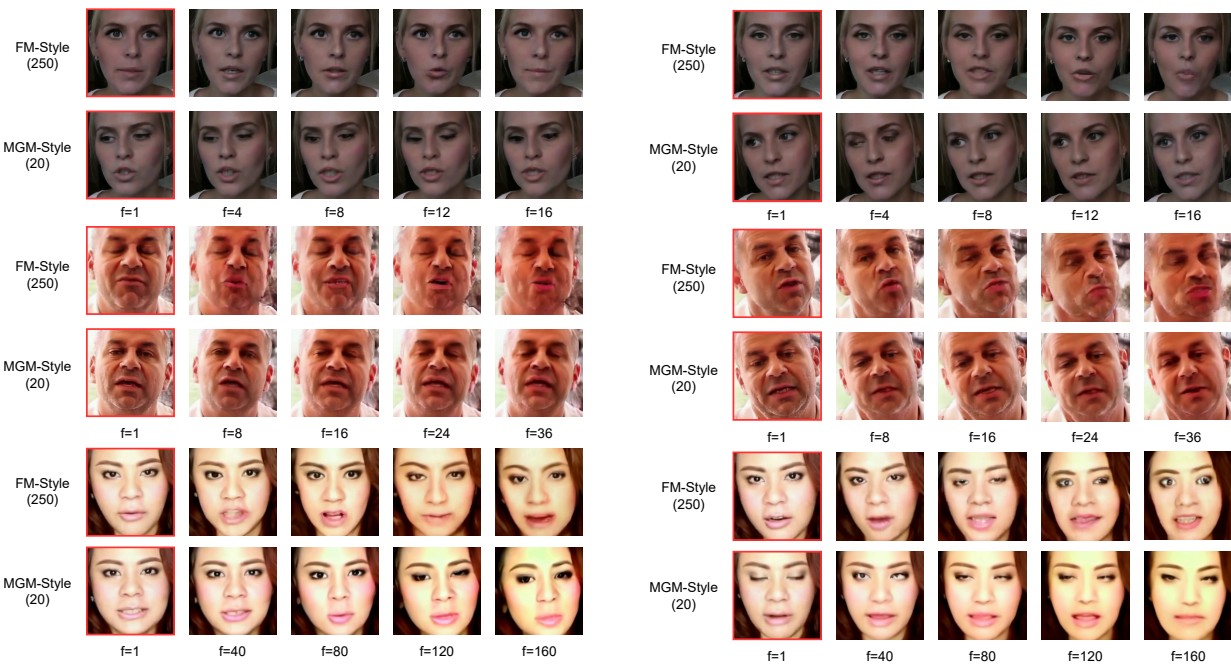

Figure S8: **Visualizations of FaceForensics generation results with different context frames.**

