# OpenReview forum: "MaskFlow: Discrete Flows For Flexible and Efficient Long Video Generation"
_TMLR — Rejected by TMLR_

### Review · Reviewer_LTFf · 2025-12-09

**Summary Of Contributions:**

This paper proposes MaskFlow, a method by which to produce long sequence videos efficiently using variable length (decidable at inference time) contexts, thus providing a sliding choice between autoregression and full-sequence video generation.  To do this, the authors combine discrete flow matching with frame-level masking and providing each frame with an independently sampled masking ratio during training. They then use MGM-style masked sampling to attempt to reduce the NFE (a proxy for runtime efficiency) while keeping long-horizon extrapolation strong.  They evaluate their method on FFS and DMLab and show FVD results relative to rolling-diffusion and diffusion-forcing baselines.

**Audience:**

Yes

**Audience Explanation:**

Should the method be scalable to larger videos and can demonstrate reasonably consistent results on their longer roll-outs, it does provide a substantially more time-efficient way to produce video.

**Claims And Evidence:**

No

**Claims Explanation:**

While I can find no significant mathematical flaws in the paper, and the proposed method is compelling, I remain unsure of the method's scalability to larger format video.  The results may support that for the examined 256x256 videos, computational efficiency can be gained without sacrificing quality on smaller short videos, but it appears as though the longest rollout tested was 360 frames (at 8 fps, this would be 45 seconds) and in the qualitative figure S7, we can already see significant artifacts appearing at f=32.

Right now, the results are only demonstrated with FVD, which although designed to measure temporal consistency, seems to only be trained on 1-10 second clips.  Additionally, FVD is not sensitive to semantic drift (the sort that we see in the top of figure S7, where the window turns green and the paint turns red), which is likely to be the core drawback with not using full-sequence video generation.

**Requested Changes:**

While I recognize the resource limitations inherent in training a video model, I think the claims of this paper warrant significantly expanded qualitative results (some of these videos could be placed anonymously on youtube).  Expanded experimental proof of quality would additionally be helpful in supporting the claims. For example, a single-blind "taste" test among a sufficient sample of participants could be used.

---

### Review · Reviewer_JbD2 · 2025-12-19

**Summary Of Contributions:**

The paper introduces MaskFlow, a framework that combines discrete representations with flow-matching to facilitate the generation of long videos. The main contribution is a frame-level masking strategy during training, which applies independent masking ratios to each frame in a video chunk. This allows the model to condition on previously generated unmasked frames at inference time, enabling rollouts ten times longer than the training window. The authors leverage Masked Generative Model (MGM)-style sampling, which utilizes confidence-based heuristics to significantly reduce the Number of Function Evaluations (NFE) compared to traditional diffusion-style sampling.

**Additional Comments:**

N.A.

**Audience:**

Yes

**Audience Explanation:**

I assume the TMLR audience is likely to be interested in this work. The long video generation is rapidly becoming a central problem in GenAI study and is still technically underexplored compared to image and short-video generation. Moreover, the ideas are conceptually close to topics TMLR readers care about (e.g., training-free sampling strategies, autoregressive versus full-sequence generation, and tokenization choices). All these make the paper relevant beyond the video domain itself.

**Broader Impact Concerns:**

N.A.

**Claims And Evidence:**

Yes

**Claims Explanation:**

The main contributions are adequately validated.

The authors claim that (1) frame-level masking in a discrete VQ token space enables flexible chunkwise and autoregressive rollout to long horizons, (2) MGM-style sampling yields substantially better NFE–FVD efficiency than diffusion-style baselines, and (3) timestep-dependent models can be sampled in a timestep-independent manner without retraining. On FaceForensics and DMLab, the reported FVD versus NFE curves, extrapolation-factor results, and ablations on masking schemes and timestep conditioning support these three points.

**Requested Changes:**

I suggest the following changes:

1, the authors should provide a more thorough and focused analysis of why discrete token spaces are better suited than continuous latents for long video generation.

2, the evaluation protocol should be updated to better reflect current practices in long video generation. For example, recent work such as LongLive incorporates VBench metrics that explicitly measure temporal consistency, motion quality, and long-range coherence, rather than relying almost exclusively on FVD.

---

### Review · Reviewer_PRwH · 2025-12-30

**Summary Of Contributions:**

This paper introduces MaskFlow, a discrete flow matching framework for efficient long video generation. By utilizing a novel frame-level masking training strategy, it enables flexible autoregressive rollouts and fast confidence-based sampling. Experiments confirm the method achieves competitive video quality with significantly fewer function evaluations.

**Audience:**

Yes

**Audience Explanation:**

Overall, the paper presents a technically sound and clearly written approach to efficient long video generation, with MaskFlow offering a well-motivated discrete flow matching framework and demonstrating notable efficiency gains over diffusion-based baselines. The proposed frame-level masking strategy and confidence-based sampling mechanism are promising and contribute to reduced computational cost with competitive short-horizon video quality.

However, several critical issues prevent a fully positive assessment. First, the baseline comparisons are potentially unfair, as MaskFlow is evaluated against discrete re-implementations of Diffusion Forcing and Rolling Diffusion rather than their original continuous formulations, making it difficult to attribute the observed gains to the proposed method rather than implementation choices. Second, the confidence-based MGM sampling mechanism lacks interpretability and diagnostic analysis; without examining which tokens are masked or prioritized, it remains unclear whether the model exhibits systematic spatial, temporal, or frequency-domain biases.

More importantly, the paper’s claims regarding long-term consistency and high-quality extrapolation are insufficiently supported. The evaluation relies heavily on coarse metrics such as FVD, without fine-grained analyses of object permanence, identity stability, or geometric drift—issues that are particularly critical for autoregressive video generation. In fact, quantitative results reveal severe degradation at long horizons (e.g., a dramatic FVD increase from 2× to 10× extrapolation), directly contradicting the claim of degradation-free long video generation. Finally, the method shows strong sensitivity to rollout hyperparameters such as stride and context window, indicating limited robustness and a reliance on careful, dataset-specific tuning.

Taken together, while MaskFlow is an interesting and potentially impactful contribution, these concerns regarding evaluation fairness, interpretability, long-term quality, and robustness substantially weaken the strength of the paper’s claims under TMLR’s evaluation criteria.

**Broader Impact Concerns:**

I do not identify any significant ethical or societal concerns specific to this work that would require an additional or expanded Broader Impact Statement. The paper focuses on methodological advances in video generation efficiency and does not introduce new application scenarios that raise distinct ethical risks beyond those commonly associated with generative models. As such, no further discussion of broader impacts is necessary.

**Claims And Evidence:**

Yes

**Claims Explanation:**

- The proposed method is well-motivated and the paper is generally easy to follow.
- The efficiency gains demonstrated by the MaskFlow are impressive compared to standard diffusion baselines.

**Requested Changes:**

- MaskFlow is compared against re-implemented discrete versions of Diffusion Forcing and Rolling Diffusion rather than their **original continuous implementations**.  The author should include comparisons with the original continuous implementations to validate true performance gains.

- The confidence-based MGM sampling is treated as a "black box" without semantic analysis. The paper fails to investigate the distribution of masked tokens, leaving it unclear if the model exhibits biases regarding spatial distribution (background vs. foreground), frequency domains (low-frequency structure vs. high-frequency texture), or temporal dynamics (static vs. moving objects).

- The evaluation relies on coarse metrics like FVD and lacks fine-grained assessments of object permanence or semantic stability. The ability of the discrete autoregressive model to prevent feature drift (e.g., identity or geometric shifts) over the claimed 10x extrapolation remains unverified, despite the authors acknowledging error accumulation as a substantial challenge.

- Contrary to claims of generating videos beyond 10x training lengths "without degradation", quantitative evidence reveals severe deterioration. For MGM-style sampling on FaceForensics, the FVD degrades drastically from 59.93 at 2x extrapolation to 214.39 at 10x extrapolation. This substantial increase indicates heavy artifacting, undermining the claim of high-quality long-term generation.

- The framework's flexibility introduces a dependency on tuning sampling parameters, specifically the stride ($s$) and context window ($m$). The results demonstrate that performance varies drastically based on these choices (e.g., $s=1$ vs $s=k-m$), suggesting that the model lacks robustness and requires careful, dataset-specific hyperparameter engineering to balance error accumulation against computational cost.

---

### Decision · Action_Editor_6Ppu · 2026-02-23

**Recommendation:** Reject

**Audience:**

Yes

**Audience Explanation:**

The topic and general direction of the work are potentially of interest to members of the TMLR audience, and the motivation behind the study is acknowledged. However, taking into account the overall evaluation of the submission, the current version does not yet meet the level of technical rigor, clarity, and empirical support required for publication in TMLR.

**Claims And Evidence:**

No

**Claims Explanation:**

After careful consideration of the paper, the official reviewer recommendations, and the authors’ responses, it is concluded that the evidence presented does not sufficiently support the claims made in the submission. Authors could have been more prompt and focussed in the rebuttal. While the paper addresses an interesting and potentially relevant problem, there are common concerns regarding the adequacy and completeness of the experimental evaluation and analysis. The results do not convincingly demonstrate the stated advantages, and key aspects of the empirical validation. Despite the clarifications provided, these issues are not sufficiently resolved to meet the evidentiary standards expected for publication in TMLR.